# Holistic Approach Promotes Failure Prevention of Smart Mining Machines Based on Bayesian Networks

**Madeleine Martinsen** [1,*] **, Amare Desalegn Fentaye** [1] **, Erik Dahlquist** [1] **and Yuanye Zhou** [2]

1   Future Energy Center, Mälardalen University, 721 23 Västerås, Sweden;
    amare.desalegn.fentaye@mdu.se (A.D.F.); erik.dahlquist@mdu.se (E.D.)
2   Baidu Inc., 100085 Beijing, China; zhouyuanye@baidu.com
*   Correspondence: madeleine.martinsen@mdu.se

**Abstract:** In the forthcoming era of fully autonomous mining, spanning from drilling operations to port logistics, novel approaches will be essential to pre-empt hazardous situations in the absence of human intervention. The progression towards complete autonomy in mining operations must have meticulous approaches and uncompromised security. By ensuring a secure transition, the mining industry can navigate the transformative shift towards autonomy while upholding the highest standards of safety and operational reliability. Experiments involving autonomous pathways for mining machinery that utilize AI for route optimization demonstrate a higher speed capacity than manually operated approaches; this translates to enhanced productivity, subsequently fostering increased production capacity to meet the rising demand for metals. Nonetheless, accelerated wear on crucial elements like tires, brakes, and bearings on mining machines has been observed. Autonomous mining processes will require smarter machines without humans that guide and support actions prior to a hazardous situation occurring. This paper will delve into a comprehensive perspective on the safety of autonomous mining machines by using Bayesian networks (BN) to detect possible hazard fires. The BN is tuned with a combination of empirical field data and laboratory data. Various faults have been recognized, and their correlation with the measurements has been established.

**Keywords:** autonomous; mining machines; artificial intelligence; bayesian networks; machine learning; predictive maintenance; smart sensing; safety





## 1. Introduction

Contemporary technologies have paved the way for digitalization, the Internet of Things (IoT), electrification, and the advent of autonomous operations. These technologies offer substantial opportunities for the industry. The mining industry faces significant challenges in meeting environmental objectives. Based on a McKinsey [1] report, the mining sector accounts for approximately 2 to 3% of worldwide $CO_2$ emissions. The predominant sources of these emissions can be attributed to diesel-powered mining equipment, electricity generation essential for production, and the supply chain and transportation aspects. The emissions' intensity across mines exhibits significant variability. For instance, in sectors such as copper and iron ore, there is a substantial twentyfold difference in the emission intensity.

In terms of energy consumption, the ventilation of underground mining accounts for around 49% [2], which corresponds to approximately 1 million tons of $CO_2$ equivalents of Sweden's annual total of approximately 50 million tons of $CO_2$ equivalents [3]. In alignment with the Paris Agreement, the Swedish government has established a target to achieve a fossil-free status by the year 2045. Simultaneously, mining corporations on a global scale have undertaken a social commitment to meet the demand for metals by 2050, a necessity to attain the specified goal.

In pursuit of this objective, mining enterprises are currently making concerted efforts to expedite the complete overhaul of their mining fleets, clearing the path for fossil-free mining.

Initiatives include transitioning to battery-powered mining machinery and electrifying segments of the transportation route. These endeavors encompass innovations like trolley lines for the sustainable conveyance of ore.

As part of their efforts to align with the goal, mining industries are undertaking measures that involve transitioning away from fossil-fuelled vehicles and embracing alternative energy sources. For instance, a shift towards the adoption of battery-electric mining machinery represents a notable step in this direction [1]. However, this transformation will not occur instantaneously, and the industry is likely to contend with a mixed fleet—comprising diesel and battery-powered vehicles, for a considerable duration.

### 1.1. Background

The shift from diesel to battery-powered mining machinery has revealed several advantages [1], including diminished maintenance costs of up to 20 to 30%. In addition, reductions in $CO_2$ emissions consequently lead to a cleaner and healthier work environment for miners [1]. This transition has the additional possibility of reducing the demand for ventilation and thereby contributing to reducing $CO_2$ emissions.

The introduction of new battery-powered mining machine fleets presents a significant challenge to the mining industry, as research in this domain remains limited. Ensuring that productivity is sustained or enhanced while upholding workplace safety becomes a critical consideration that the industry must address. Illustrative instances of risks associated with battery-powered mining machines encompass concerns such as fire outbreaks, the generation and management of smoke, and fire suppression strategies [4]. Additionally, challenges include issues related to ventilation, evacuation protocols, and the release of gases, which are not solely flammable but can also possess toxicity [5]. Commercial Li-ion cells contain sources that can release toxic fluorine gases such as hydrogen fluoride (HF) and other harmful gases if undergoing failures [6,7].

A recent study [8] concerning fires on electric vehicles (EVs) compared to internal combustion engine vehicles (ICEVs) states that statistics on incidents in enclosed spaces (like tunnels and enclosed parking garages) for EVs are currently very limited. This is equally applicable to mining machines, emphasizing the importance of early fire detection through fixed detection systems and/or the integration of intelligent systems in vehicles.

### 1.2. Fires

In today's mining, heavy-duty vehicles and machines are subject to various potential hazards that can lead to fire [5,9]. Hazards such as oil leaks, engine overheating, cable reel occurrences, cable overheating, and brake-oil leakage can cause fire and thereby cause downtime, financial losses, and safety risks. Preventive measures such as regular maintenance, inspections (described in Appendix A), and appropriate safety measures are ways to avoid these hazards from occurring and ensure the safe and efficient operation of mining heavy-duty vehicles and machines. The major faults on a mining machine causing fires in the mining are summarized in Appendix B.

Concerning fires in underground hard rock mines, statistical studies show resembling conditions and patterns the world around. For metal and non-metal mining [10], the most similar types of equipment involved in fires were vehicles and mobile equipment, oxyfuel torches, beltlines, electrical systems, batteries, chargers, heaters, cutting saws, explosive boxes, and air compressors. The cause of fires [11,12] proved to be hydraulic fluid, oil or fuel sprayed onto hot surfaces, hot works, and electrical shorts or arcing.

The types of fuel involved in the fires followed the dominating fire causes, with the most frequent fuel types being hydraulic fluid, oil or fuel, electrical cords, cables, wires, batteries, and oxyfuel, clothing, or grease. The predominant factors believed to contribute significantly to hydraulic failures include hydraulic leakage and pipe (hose) ruptures [5,13].

Among the most prevalent fire occurrences, mining vehicles, including service vehicles, drilling rigs, and loaders, were found to be the most common objects involved. Statistically,

fire incidents in Swedish underground mining transpire approximately once a week [14,15], with 80% of these occurrences attributed to mining vehicles and machinery [16].

### 1.3. Aims and Objectives

In the context of autonomous mining operations characterized by limited or no human intervention in production processes, there arises an imperative to enhance methods for pre-emptively detecting, for instance, maintenance requirements. Artificial intelligence (AI) is predicted to play a significant role in this transformation [17].

Today's different maintenance approaches are applicable in many industries [18] and can be recognized as corrective, preventive, and predictive maintenance. Corrective maintenance is performed in response to issues or malfunctions, while preventive maintenance means inspections and maintenance performed after a fixed plan. Predictive maintenance depends on the condition of assets and is a way to predict the future condition of an asset by estimating when it will deteriorate from its current condition.

In an illustrative instance involving the utilization of AI to forecast the source of fires within tunnels [19], the trained model achieved an impressive accuracy rate of 90%. Recent articles and studies in the field of machine and fire safety have predominantly focused on health aspects [20], accident prevention for machines [21], and the operator perspective [22].

It is essential to have the capability to detect underlying causes as well as necessary preventive measures; from these, accurate predictions can be established. According to [23], machine learning (ML) output seems to lack the capabilities to handle uncertainties and reasoning skills to assess causal relationships, which is why BN is advocated. The study [24] shows an example where a BN model was introduced to meet the challenges concerning unused data for maintenance decision support to overcome the limitations of statistical models. This model proved successful in extending an existing statistical model of asset deterioration.

Unfortunately, there is a notable scarcity of research when it comes to addressing fire safety and implementing smoke detection measures on underground mining machines. This highlights an interesting challenge and opportunity for developing smart machines. The novelty of this research lies in the automation of maintenance decision-making through the utilization of machine learning methods. Moreover, the capability to predict fire-causing hazards at the earliest feasible juncture and provide suitable protective measures is a distinctive contribution. The objective of this study is to augment existing detection systems by pinpointing risks stemming from material degradation in mining machinery. This effort seeks to provide timely support and alerts as necessary. To address this challenge, it becomes imperative to deploy methodologies capable of detection, and from initial studies [16,20,25], a range of diverse sensors, such as gas sensors, temperature sensors, and FLIR cameras, were evaluated and tested. This paper focuses on the error detection system and an intelligent decision support mechanism, which could be essential to be able to provide recommendations for appropriate actions of autonomous mining machines.

## 2. Methodology

Decomposition of materials is a factor that poses risks for fires and production losses but also causes health problems. In previous studies, the methodology focused on how to measure the degradation of products and heat release rate [26]. Whilst this article leverages the aforementioned knowledge to craft a comprehensive methodology for detecting various types of faults in mining machinery and formulating strategies to effectively mitigate their impact, in doing so, it endeavors to proactively prevent the occurrence of potentially severe outcomes. From online measurement, experimental data [26], and domain knowledge, a smart decision support tool can be developed.

### 2.1. Degradation of Materials and Measurements

Overheated plastic material from cables, motors, and the like, oil from hydraulic and brake systems, electrolytes from batteries, and rubber from tires, for instance, will

all emit different types of organic compounds. Different levels of toxicity exist in these substances, some of which are harmful to human health. Acids produced, for example, by battery electrolytes and halogen-containing polymers like polyvinyl chloride (PVC) and Teflon, polytetrafluoroethylene (PTFE) also can cause significant long-term problems for production equipment and especially electronics, as products like hydrochloric acid (HCl) and hydrogen fluoride (HF), where HF is produced by thermal degradation of the plastics.

Polyacrylamide can produce hydrogen cyanide (HCN), which is extremely toxic in very low concentrations. Other gases that can be released by mining machines come from poor combustion in engines as well as pyrolysis of plastic, which can produce carbon monoxide (CO) and nitrogen oxides ($NO_x$). Different and more complex compounds, like polycyclic aromatic hydrocarbons (PAH) and aromatics containing nitrogen, sulphur, or halogens [27,28], may be produced, and, in addition, CO and $NO_x$ are produced during blasting in the mine, both in an open-pit as well as underground, and need to be ventilated away before underground miners enter the area or manually operated trucks start to transport the ore in the open pit.

In Sweden, the permitted limit of CO is 20 ppm long-term (8 h average exposure time for miners), 100 ppm short-term (15 min exposure time for miners), and for $NO_2$, 1 and 5 ppm, respectively [29]. Despite the introduction of autonomous vehicles, the issue of producing toxic substances necessitates attention, particularly because individuals will be responsible for servicing mining machinery and conducting repairs in the event of breakdowns within the mining environment. Other harmful airborne particles [20] that a miner at an open pit can be exposed to are mineral dust (including crystalline silica), acid mists, airborne solvents, and also high noise levels, high vibration levels, and extreme temperatures. Symptoms observed [20] among miners appear to be shortness of breath, wheezing, coughing, and increased sputum production.

### 2.2. Smart Failure Prevention

Numerous industrial sectors have leveraged AI applications, including BN, in several fields, such as aviation [30], transportation [31], and manufacturing [32]. AI methodologies can be categorized into three distinct classes, as depicted in Figure 1: "discover unknown principles with data"; "modeling industry process with known principles and small amount of data"; and "modeling industry process with large amount of data". "Discover unknown principles from data" is mainly related to the symbolic regression, for example, Brunton et al. [33] proposed a sparse identification algorithm (called SINDy) to reconstruct the Navier–Stokes equation from flow velocity data, and the SINDy algorithm was further applied to find the governing equation of plasma flow in a Tokamak nuclear fusion device, and hence, it was used to provide control guidance [34]. "Modelling industry process with known principles and small amount of data" is mainly related to the physics-informed neural network (PINN) proposed by Raiss et al. [35], which demonstrates commendable performance in both scenarios—the forward and inverse problem of partial differential equations [36–38]. Nevertheless, the final category represents the AI method that has gained the widest application, "modeling industry processes with large amount of data". This class is also called a "data-driven AI model", as it only relies on the data when building models without requiring a prior understanding of underlying principles.

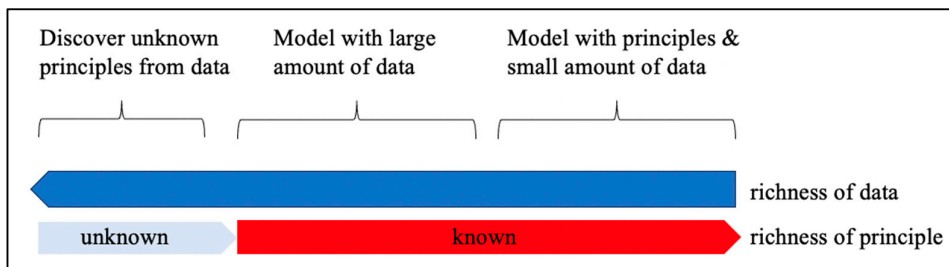

**Figure 1.** Classification of three AI methods.

Some renowned data-driven AI models include neural network models, such as transformer neural networks [39], graph neural networks (GNN) [40], and convolutional neural network [41]; other models, such as support vector machines [42], Bayesian networks [43] and decision trees [44]. It is worth acknowledging that neural network models are widely embraced across various domains due to their straightforward architecture and efficacy. However, these models also encounter challenges, including insufficient data, overfitting [45], and explainability [46]. As a result, addressing the challenges posed by neural networks has led to a preference for alternative models.

This study utilizes the BN to construct an AI model aimed at pre-emptive failure prevention in mining machinery. It represents a model type characterized by its efficacy with limited data. The rationale behind selecting the BN lies in the scarcity of failure event data for mining machinery and the need for transparency in the AI model to facilitate tracking failure causes.

The BN is developed based on Bayes' theorem and is a graphical model that consists of nodes and edges [47]. The nodes describe the events and the edges describe the relationship between two events. The probability of an event is assigned to its corresponding node; therefore, the posterior probability of an event under certain conditions can be inferred from the BN.

BNs demand a smaller volume of data compared to neural networks, alleviating concerns regarding overfitting. Moreover, their enhanced explainability sets them apart. So far, it has achieved excellent performance in safety and risk management, as described in the literature [48–52]. Regarding the domain of AI related to mining safety, the majority of studies tend to focus on gas explosions. For example, Li and Fang [53] analyzed the cause of gas explosions based on BN and identified the main cause; Li et al. [54] combined BN with a fuzzy analytic hierarchy process for gas explosion risk assessment. Unfortunately, there has been limited research conducted in the area of mining machine safety.

*2.3. Bayes' Theorem*

Bayes' theorem articulates the probability of an event in relation to specific conditions, as follows:

$$P(A|B) = P(B|A)\frac{P(A)}{P(B)} \tag{1}$$

where $P(A|B)$ is the posterior probability of A given B, $P(B|A)$ is the posterior probability of B given A, $P(A)$ and $P(B)$ are the prior probability of A and B, respectively. Posterior probability is also called likelihood; prior probability is also called marginal probability. Table 1 shows an example of how BN was applied for machine failure: hydraulic leakage.

**Table 1.** BN calculation for hydraulic oil leakage.

| Conditional Events (P) | Probability |
|---|---|
| $CH_x$: High | 0.0198 |
| $CH_x$: Low | 0.9802 |
| HydraulicLeakage: Yes | 0.01 |
| HydraulicLeakage: No | 0.99 |
| $CH_x$: High \| HydraulicLeakage: Yes | 0.9 |
| $CH_x$: Low \| HydraulicLeakage: Yes | 0.1 |
| $CH_x$: High \| HydraulicLeakage: No | 0.01 |
| $CH_x$ Low \| HydraulicLeakage: No | 0.99 |

Based on historical statistics and internal data, the probability of no hydraulic leakage occurring is 0.99, and the probability of the $CH_x$ level being high is 0.0198 under the conditions. Once the probability of the $CH_x$ level with given hydraulic leakage conditions

is determined, it is possible to obtain the probability of hydraulic leakage with given $CH_x$ levels:

$$P(HydraulicLeakage : Yes|CHx : High) =$$
$$P(CHx : High|HydraulicLeakage : Yes)\frac{P(HydraulicLeakage: Yes)}{P(CHx: High)} \quad (2)$$
$$= 0.9 * \frac{0.01}{0.0198} = 0.455.$$

In this specific case, it was observed that when the $CH_x$ level was elevated, the associated probability of hydraulic leakage reached 0.454. For a more complicated system, a BN was used. However, the fundamental principle was Bayes' theorem.

### 2.4. Hazard Detection Framework

A holistic fire hazard detection framework based on BN for mining machines is shown in Figure 2. It is an integrated system that uses sets of trained BN to detect hazards in mining heavy-duty vehicles that might lead to fire, and then provides decision support based on the severity of the problem.

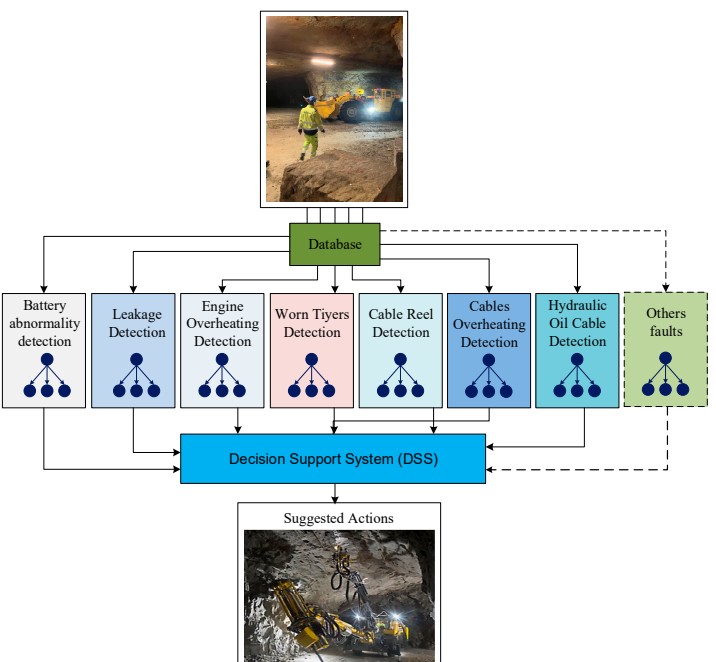

**Figure 2.** Hazard detection framework in mining heavy-duty vehicles.

The framework uses different measurements from various types of sensors [5] consisting of both electrochemical sensors (to capture CO, $NO_x$, and $SO_2$), photoionization (Hydrocarbons), and catalytic and IR sensors ($CH_4$, $CH_x$, and $CO_2$). Further, temperature, pressure, and airflow measurements are installed at different locations in the mining area and on heavy-duty vehicles, which continuously measure the relevant parameters and provide input data for the BN models. Multiple BN modules have been developed to detect, for instance, fire-causing faults in mining areas, including engine overheating, cable reel, hydraulic leakage, brake leakage, cable overheating, worn tires, and battery abnormality. Each BN module is trained using historical data collected through the sensors. The output of the BN modules is a set of probabilities representing the likelihood of a particular fault occurring.

The output information from the BN modules is used as input for the decision support system. The decision support system processes the output data and generates recommendations for actions that can be taken to mitigate the fire hazard risks.

The decision support system also provides alerts and suggested predictive measures to the automated maintenance system in real time if a fire hazard is detected. The procedure used to develop the detection modules corresponds to the following steps:

- Identify the variables and define their states.

As shown in Figure 3, measurement parameters that are relevant to detecting the identified fire-causing hazards are selected. Tire wear, break leakage, cable reel, engine, and cable overheating are manifested by measurement of the variable's temperature, pressure, and $CH_x$ deviations due to degraded polymers, from their normal operational value. For measurements, sensors are installed around and on the heavy-duty vehicles. To detect hydraulic leakage and brake leakage, oil level and temperature variation are considered as evidence. The definition of states and normal level is determined from expert experience and experimental results [5,26]. For any $CH_x$ value above 0 ppm, it is a risk, where the states are defined to be either low, medium, or high. However, for temperature, no 0 level exists, and for that reason four states are defined (OK, low, medium, high).

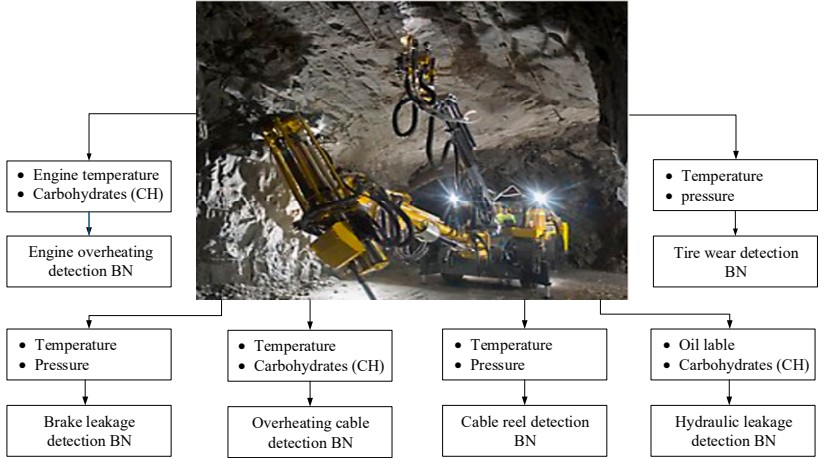

**Figure 3.** Desired measurements to detect each fire causing faults.

- Construct the BN.

The BN structure represents the connection between fire-causing hazards (for example, engine overheating, cable overheating, etc.) and the evidence (changes in temperature, pressure, $CH_x$, and oil label measurements). Each fire-causing hazard is the parent node, while the associated measurement changes represent the child nodes. As illustrated in Figure 4, the states for the parent nodes are defined based on the nature of the problem and its relevance in the decision-making process. The states for the evidence, on the other hand, are defined by sorting measurement deviations and dividing them into non-overlapping intervals. These states are described linguistically as YES, NO, OK, low (L), medium (M), and high (H).

As an example, for cable overheating detection BN modules, the OK state refers to temperature conditions below 50 °C, requiring $CH_x$ levels to be maintained at less than 0.1 ppm. For the L state, temperatures ranging from 50 °C to 65 °C are considered, accompanied by $CH_x$ levels that fall between 0.1 ppm and 0.7 ppm. These initial states emphasize the significance of keeping $CH_x$ presence minimal within lower temperature ranges. Temperatures between 65 °C and 100 °C fall under the M state. $CH_x$ levels within this range are expected to be maintained between 0.7 ppm and 1.2 ppm. The H state encompasses temperatures exceeding 100 °C, where $CH_x$ levels are required to be above 1.2 ppm. All data have been retrieved from presented experimental data [26].

- Determine prior probabilities.

The prior probabilities of the variables were determined based on historical data and expert knowledge. It is important to note that prior probabilities are not always available or accurate, and they may need to be updated based on new evidence. Data have been acquired from the Swedish underground mining fire experience [9,14], shown in Figure 5, but also internal data provided by machine vendors and mining businesses for expert

knowledge. Based on this knowledge, the mining machines' engines and other fire-causing accessories or equipment are assumed to be operating without failure for 99 percent of their life span.

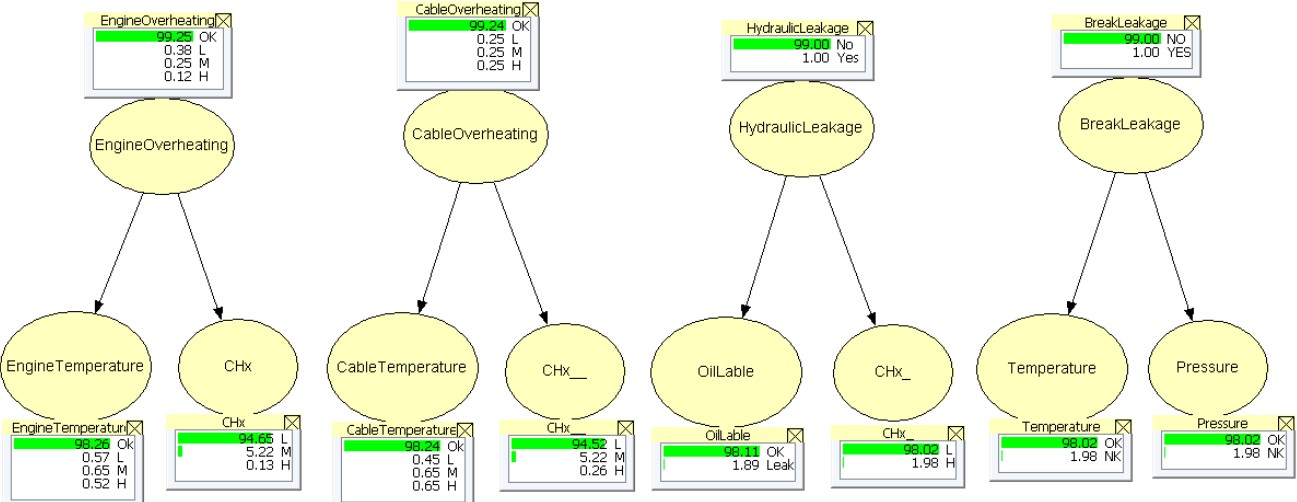

**Figure 4.** Statistics of parent nodes and child nodes.

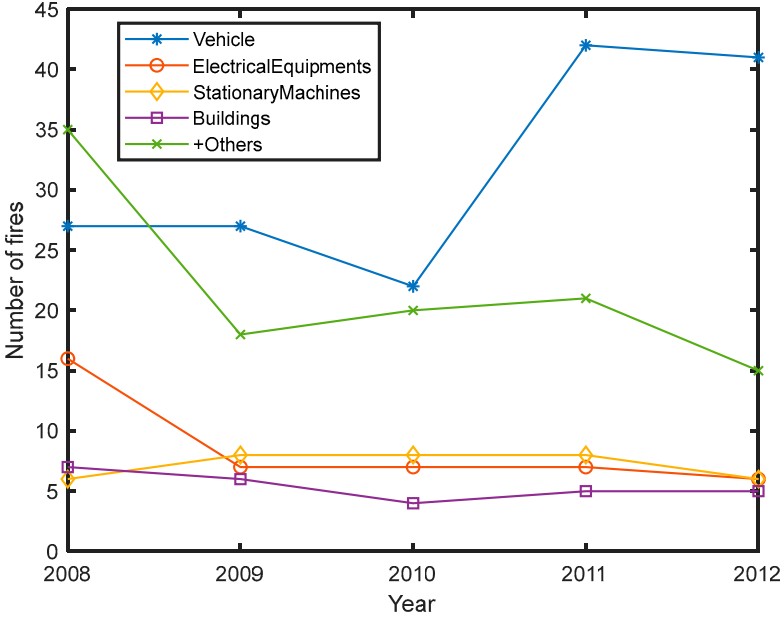

**Figure 5.** Fire events in Swedish mines observed from year 2008 to 2012.

- Compute conditional probability tables (CPTs).

CPTs define the relationship between parent and child nodes or the fire-causing faults and evidence/measurement deviations. The CPTs for the proposed structure were assigned based on historical data and expert knowledge. An example of CPT for the engine overheating case is shown in Table 2.

**Table 2.** CPT for the engine overheating BN module states.

| Parent States | Engine Temperature | | | | CH$_x$ Measurement | | |
|---|---|---|---|---|---|---|---|
| | OK | L | M | H | L | M | H |
| OK | 0.99 | 0.002 | 0.004 | 0.004 | 0.95 | 0.05 | 0 |
| L | 0 | 0.98 | 0.02 | 00 | 0.95 | 0.05 | 0 |
| M | 0 | 0.015 | 0.97 | 0.015 | 0 | 0.95 | 0.05 |
| H | 0 | 0 | 0.01 | 0.99 | 0 | 0.01 | 0.99 |

- Infer the state of the system.

The posterior probability values are calculated from the trained BNs by inferencing the measurement deviations. The Bayes' theorem is applied using inference algorithms to estimate the posterior probabilities. Interpreting the probability values obtained through the inference process was performed for the detection of the corresponding fault based on defined probability thresholds.

Fault detection is determined if the sum of the estimated posterior probability of the parent states other than the OK state is greater than 0.5. Additionally, the state with the greatest probability is considered as the severity level of the problem. For instance, if the estimated posterior probability of state M is higher than the remaining states' posterior probability values, the severity of the detected fire-causing fault is medium. For example, with cable overheating, different cable materials can withstand different temperatures without degradation. If the sensors indicate both increased temperature and increased CH$_x$ level, it is a strong indication of overheated cables with potential risk for fire through time. If the sensors indicate only increased temperature or increased CH$_x$ level, the probability is lower for a hazardous situation. Experiments were performed with different types of cables with respect to insulation and capacity [26]. The probability values in the BN are fed with this experimental data. The predictive probability values in the BN are also verifying the evidence.

- Implementation of the BN model.

The BN model will be tested by using a new dataset from machines in operation to evaluate its performance in detecting faults. This phase is ongoing and will demand several months of testing to retrieve relevant data and make statistical analyses. Furthermore, fine-tuning of the BN model parameters will be carried out. From the first experiments on a single machine, initial data was retrieved. However, the entire fleet is now in focus.

A complete example of the engine overheating detection BN module has been provided in Appendix C. This demonstrates the entire procedure, from identifying the variables to defining their states until training and testing the model. In this example, the following points have been addressed:

- Why and how the variables were selected;
- Why and how the states of each variable were determined;
- How the relevant probabilities were determined and verified;
- How the model was verified.

Future research endeavors aim to outline potential trajectories for autonomous mining machines, given their financial parameters. For instance, if a considerably aged mining machine exhibits engine issues, directing the autonomous mining machine to the scrapyard might be a more prudent course of action compared to routing it to a workshop.

## 3. Results and Discussion

### 3.1. Measured Data from Sensor

Data collection will be facilitated through an array of sensors, encompassing fire, smoke, CO, NO, NO$_2$, hydrocarbon, heat, and FLIR sensors, affixed to both mining machines and mining walls. This collective sensor data forms the fundamental input for the

safety system. This information is then transmitted to a BN, suggested in this paper, which systematically traces the path to the source and evaluates potential risk scenarios.

The foremost objective of the sensors will be to detect potentially hazardous situations that underground miners could encounter. In Figure 6, an example of acquired data for one day in year 2021 and year 2022 is shown. The data are from the same hours in the morning but different days and years. From this field test, notable findings of a $CO_2$ sensor mounted on a diesel mining machine registered elevated $CO_2$ levels. $CO_2$ levels between 3000–5000 ppm of exposure are highly likely to induce dizziness in the machine operator, consequently impacting their reaction capabilities.

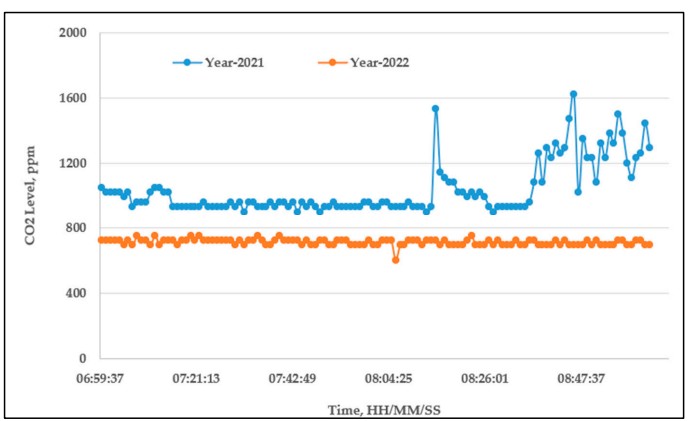

**Figure 6.** $CO_2$ level measured on a underground mining machines (diesel).

Consequently, this circumstance can also exert an influence on miners operating in the immediate vicinity of the machine. These measurements were derived from a single machine, raising the question of how air quality might be impacted and what levels of $CO_2$ emissions underground miners could potentially be exposed to when multiple machines are operational within the same shaft.

One pertinent concern pertains to the calibration of the sensors. During the testing phase, this issue was addressed by involving the service personnel to recalibrate the sensor whenever maintenance procedures were performed on the machine. A noteworthy finding during the testing phase was the resilience of the $CO_2$ sensor to moisture and dirt, both prevalent factors within mining environments.

Earlier results obtained from a sequence of tests involving signal collection from sensors mounted on mining machinery [16,26] and even on unmanned aerial vehicles (UAVs or drones) [50] suggest a viable potential to acquire meaningful data, enabling proactive measures against hazardous situations, and potential threats to the well-being of mining personnel. These datasets have been employed in the formulation and execution of BN calculations in this paper.

Another illustrative outcome of this research and conducted tests at an open pit—Aitik (Boliden)—to observe the dynamics between gases and sensors is the transformation of once-hazardous tasks, such as post-blasting gas measurements, into fully automated processes. Previously, such tasks required manual intervention, with a lead engineer driving a vehicle in the open pit, manually lowering the window of the car to measure, with a hand-held sensor, explosive gases like carbon monoxide (CO), nitric oxide (NO), and nitrogen dioxide ($NO_2$). If the readings were outside acceptable limits, the vehicle would be reversed, and the procedure would be repeated until suitable values were attained before mining trucks could proceed to collect the ore. Today, this process has evolved significantly, as an operator situated in the control room initiates a drone inspection and gas measurement with a single button press. The drone is programmed to navigate the open pit independently, conducting meticulous gas measurements.

Accordingly, a comprehensive sensor framework is proposed to further utilize sensor data, as shown in Figure 7. The system incorporates sensors positioned on mining walls,

vehicles, individuals' clothing, helmets, drones, and various other sources, generating a multitude of signals. These signals can serve to ascertain the severity of issues and, furthermore, to pinpoint the problems as proximately as feasible to their origins. Based on these insights, both a model and diagnostic systems are developed. Consequently, this paves the way for forthcoming opportunities to simulate diverse scenarios encompassing potential processes. This approach facilitates a holistic comprehension of the potential ramifications and aids in conducting risk assessments concerning the varied measures implemented.

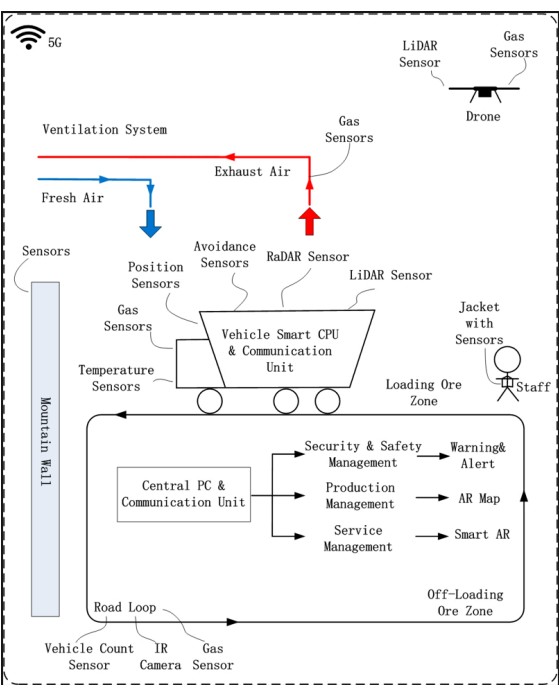

**Figure 7.** Sensor-based approaches enhancing safety in autonomous underground mining.

### 3.2. Simulated Data

In mines, the measurement data are usually sparse and may not be sufficient for the BN. For example, having a gas source, as marked in Figure 8, the sensor value left of the source (marked with a square) will be lower than the sensor value close to the source. Therefore, simulation data are generated to meet the requirement.

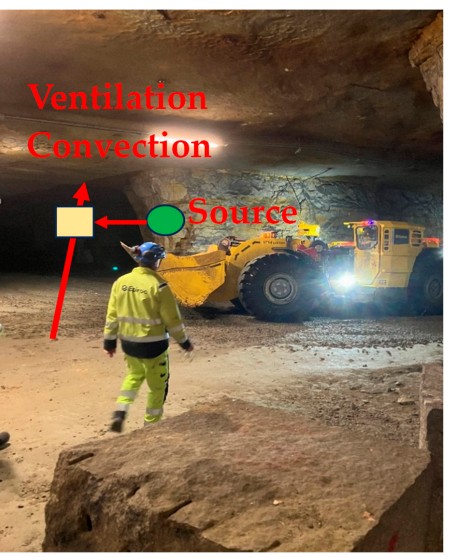

**Figure 8.** Diffusion of gas concentrations.

Assume a source emitting gases, for instance, an overheated cable near a rock surface. The primary gas emission is $CH_x$, although the emission of other gases may also occur, contingent upon the material of the cable. These gases slowly diffuse into the tunnel, but due to ventilation and passing vehicles, it is mixed into the air and diluted. By simulating what the concentration profile will look like from the source to any position in the tunnel, knowing the flow pattern of ventilation air or the turbulence flows created by a vehicle passing the position. The concentrations can be predicted at any position. This can be simulated generally by Equation (3) below, where C is the concentration of gas components, for example, $CH_x$ at time x is the spatial position, D is the diffusion constant, and $v_x$ is the convection coefficient in x direction. In practice, it is essential to account for both the $v_z$ and $v_y$ directions.

$$\frac{dC}{dt} = -D\frac{d^2C}{dx^2} + v_x\frac{dC}{dx} \tag{3}$$

Normally, convection is much more influential than diffusion. From the simulation, we can approximately determine what the concentration should be at a certain distance from the source depending on the ventilation flow and the impact of the vehicles passing by. If we are far away from the source, a low concentration measured can still indicate a high probability of a risk for fire.

An example of a simulated concentration of $CH_x$ at different positions is shown in Figure 9, assuming there is a peak generation of $CH_x$ at position C0 at time t = 1 until time t = 12. For example, the concentration in the square in Figure 8 corresponds to position C6 in Figure 9.

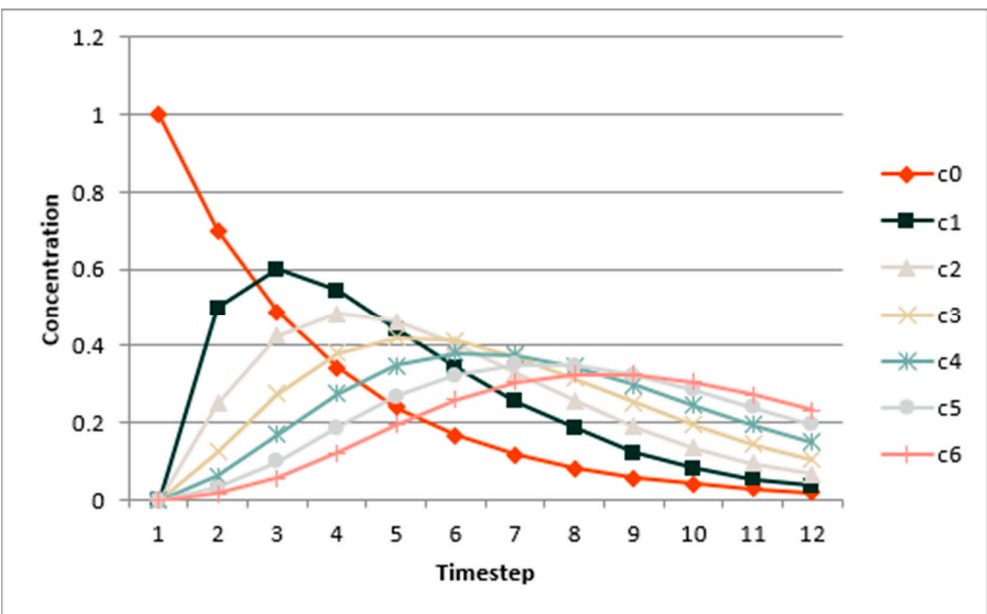

**Figure 9.** Concentration of $CH_x$ (*Y*-axis) from the source C0 at the wall out into different positions in the tunnel. C0–C6 represents different positions from timestep = 1 to 12 (*X*-axis). No fixed dimension.

This is a general solution being shown in Figure 9 and not the absolute values. Timestep could be in seconds while concentration, C, could be ppm. Position 0 is the source, and the concentration varies depending on if overheating is a fact or not. In volume element 0, the concentration is c0. For instance, in timestep 2, the concentration in volume element 1 was 0.5. Equation (3) shows how it can be simulated. The value in a certain position depends on the effect of ventilation, vehicle movement, and the behavior of the source. This is addressed by using a fixed ratio between $\Delta t/\Delta x^2$. Assuming scaling up $\Delta x$ from 1 dm to 3 dm, the value of $\Delta t$ increases correspondingly.

*3.3. BN Results*

Measured data, like fault signatures collected from field data [16,25,26], combined with simulated data for the determination of overall risks using the BN constituted the base when testing the BN model.

As illustrated in Figure 10, inference algorithms are used to estimate posterior probabilities based on the Bayes' theorem. If the estimated posterior probability for any of the parent states, other than the state "OK", is above 50%, the system or component where that state belongs to is faulty. The detected state (L, M, or H) refers to the severity of the fault. If the posterior probability for the "OK" state shows over 50%, the system is considered as "healthy", and no maintenance action is needed. It is also relevant to consider the density of different gases. Nonetheless, our current approach has entailed qualitative considerations rather than quantitative assessments.

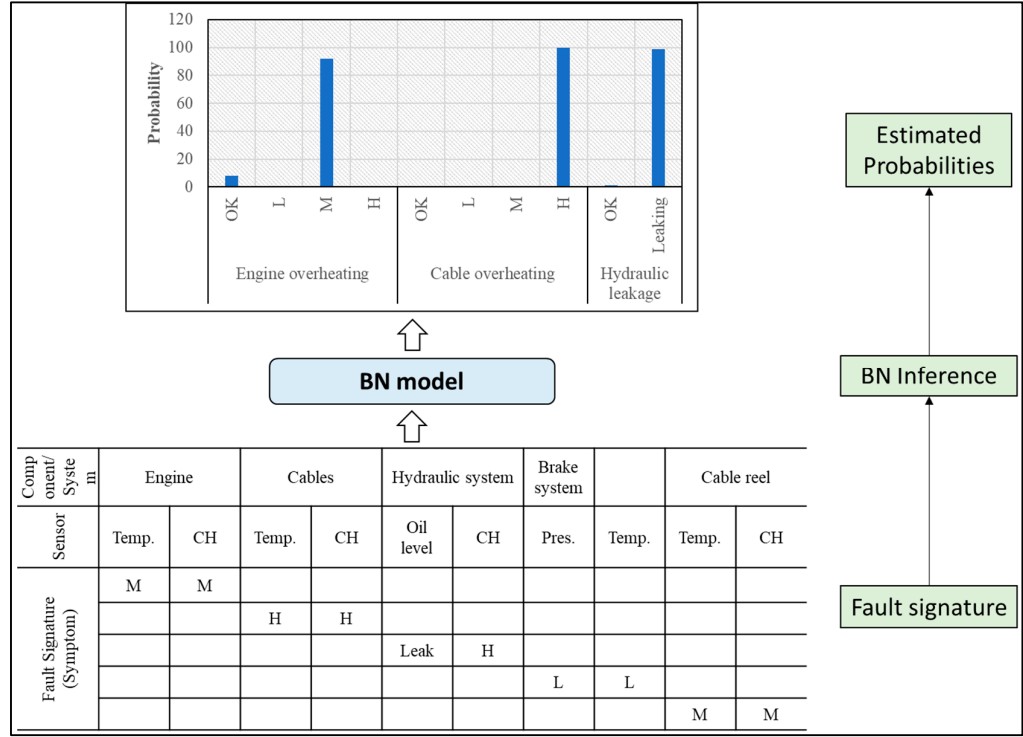

**Figure 10.** Testing the BN model.

Tables 3 and 4 show the possible fault signatures and their corresponding posterior probabilities of the hazard states (parent node states), respectively. The estimated probability values presented in Table 4 indicate the overall detection accuracy of the proposed BN scheme. The twelve cases refer to the number of parent states considered associated with the four possible fire-causing objects ("Engine overheating", "Cable overheating", "Hydraulic leakage" and "Brake leakage").

In Table 3, the measurement signatures considered in cases 2, 3, 4, 6, 7, 8, 10, and 12 indicate the occurrence of a fault, and the corresponding estimated probabilities presented in Table 4 confirm the existence of a fault in these cases. The remaining test cases (case 1, case 5, case 9, and case 11) in Table 3 represent situations with no faults. The estimated probabilities provided in Table 4 also affirm the absence of any faults in these cases.

According to Figure 4, the child nodes of the BN model for engine overheating detection can have four possible combinations: OK/L, L/L, M/M, and H/H. The same applies to the cable overheating BN model. For the hydraulic leakage detection BN model and the Brake leakage detection BN model, two cases can be drawn for each: OK/OK and NK/NK. In total, there are 12 possible fault signature combinations for inference. It is evident that the BN modules exhibit a high degree of accuracy in detecting the specific

faults associated with fire causation. From experiments [26], by overheating cables with high currents, knowledge was gained by subjecting cables to high currents and overheating them, determining the temperatures at which they emitted gases like $CH_x$. When the temperature remains within normal operational ranges and the $CH_x$ sensor indicates a low ppm reading, the likelihood of detection of faults is negligible, as exemplified in case 1 of Tables 3 and 4.

**Table 3.** Example fault signatures considered in the BN test.

| | Engine Overheating | | | | | | | Cable Overheating | | | | | | | Hydraulic Leakage | | | | Brake Leakage | | | |
| | [1] Temp. | | | | $CH_x$ | | | Temp. | | | | $CH_x$ | | | Oil Level | | $CH_x$ | | Pressure | | [1] Temp. | |
| Case | OK | L | M | H | L | M | H | OK | L | M | H | L | M | H | OK | Leak | L | H | OK | [2] NK | OK | [2] NK |
| 1 | x | | | | x | | | | | | | | | | | | | | | | | |
| 2 | | x | | | x | | | | | | | | | | | | | | | | | |
| 3 | | | x | | | x | | | | | | | | | | | | | | | | |
| 4 | | | | x | | | x | | | | | | | | | | | | | | | |
| 5 | | | | | | | | x | | | | x | | | | | | | | | | |
| 6 | | | | | | | | | x | | | x | | | | | | | | | | |
| 7 | | | | | | | | | | x | | | x | | | | | | | | | |
| 8 | | | | | | | | | | | x | | | x | | | | | | | | |
| 9 | | | | | | | | | | | | | | | x | | x | | | | | |
| 10 | | | | | | | | | | | | | | | | x | | x | | | | |
| 11 | | | | | | | | | | | | | | | | | | | x | | x | |
| 12 | | | | | | | | | | | | | | | | | | | | x | | x |

[1] Temp means temperature. [2] NK means 'NOT OK. Cases 1–12 shows examples of fault scenarios.

**Table 4.** Estimated posterior probability values of the parent node states (in %) for the given conditions in Table 3.

| | Engine Overheating | | | | Cable Overheating | | | | Hydraulic Leakage | | Brake Leakage | |
| Case | OK | L | M | H | OK | L | M | H | NO | YES | NO | YES |
| 1 | 100 | 0 | 0 | 0 | | | | | | | | |
| 2 | 34.95 | 65.05 | 0 | 0 | | | | | | | | |
| 3 | 7.88 | 0.16 | 91.96 | 0 | | | | | | | | |
| 4 | 0 | 0 | 0.16 | 99.84 | | | | | | | | |
| 5 | | | | | 100 | 0 | 0 | 0 | | | | |
| 6 | | | | | 44.32 | 55.68 | 0 | 0 | | | | |
| 7 | | | | | 7.8 | 0.11 | 92.09 | 0 | | | | |
| 8 | | | | | | | 0.08 | 99.92 | | | | |
| 9 | | | | | | | | | 100 | 0 | | |
| 10 | | | | | | | | | 1.1 | 98.9 | | |
| 11 | | | | | | | | | | | 100 | 0 |
| 12 | | | | | | | | | | | 1 | 99 |

Cases 1–12 shows examples of fault scenarios.

Nevertheless, in instances where a low $CH_x$ reading is received, it could potentially be attributed to airflow patterns and, hence, escape sensor detection. Elevating sensor

sensitivity stands as a pivotal means of reducing this uncertainty, thereby addressing a substantial sensitivity concern. If temperature is the only measurement available, the same temperature can be serious for one type of cable insulation while not for another, as shown in cases 5 to 8 of Table 3. However, in the table, an example of a single cable type is presented. These insights form the basis for maintenance decision-making recommendations that inform the actions of the autonomous mining machine's operator or management system through the employment of a BN model. For instance, as seen in Table 4 from case 10, wherein hydraulic oil leakage detection is considered, the BN model showed a 98.9% estimated posterior probability for leakages occurring under conditions with high levels of $CH_x$, as shown in Table 3 for case 10.

For maintenance purposes, detecting brake leakage at any level is sufficient for measurement and appropriate action. In the case of leakage, the machine requires immediate intervention. However, when it comes to engine and cable overheating, the severity of the fault plays a crucial role in determining the course of action. In instances of low and medium-level engine and cable overheating, immediate action may not be necessary, and the machine can continue its operation. Nevertheless, this serves as an indication and should be reported for subsequent maintenance checks.

## 4. Conclusions

Canary birds, known for their small size, have historically served as "sensors" in underground mines, being utilized to detect and alert miners to the presence of hazardous air conditions. When they stopped chirping, it was a warning to the miners that the air quality was poor. Whereas if the bird lay on the bottom of the birdcage, this constituted an alarm to immediately evacuate the mine. This research investigates the potential feasibility of replacing the canary's rapid response and sensitivity to hazardous air conditions with sensors, thus serving as a supplementary component within the safety system.

In forthcoming autonomous mining operations within underground environments, human involvement will remain crucial for maintenance and servicing tasks. The implementation of sensor-based tracking systems to monitor airflow and airborne particles offers the potential to provide early alerts for non-respirable atmospheric conditions, enhancing the safety of maintenance operations. Moreover, different sensors can play a pivotal role in assessing the operational health of machines that, for instance, emit gases (emission as well as degradation of plastics or oil leakage), thereby detecting potential malfunctions that may escalate fire hazards. In addition, measurement of temperature is utilized.

A holistic fire hazard detection framework based on BN for mining machines has been explored. Furthermore, this paper has showcased a comprehensive approach aimed at enhancing failure prevention in intelligent mining machinery. The procedure used to develop the detection modules was presented. The methodology employed BN to address some of the prevalent fire-inducing failure scenarios. The BN was tuned with a combination of empirical field data, laboratory data, and expert knowledge.

The progression of autonomous vehicle development, for instance, Epiroc's drilling rig and Volvo's development of automated dumpers, is already in motion. It is probable that we will witness the emergence of intelligent mining machinery and equipment functioning underground, inclusive of adept inspection protocols and on-demand maintenance services, in the relatively near future.

**Author Contributions:** Conceptualization, M.M., A.D.F., Y.Z. and E.D.; methodology, M.M., A.D.F., Y.Z. and E.D.; software, M.M. and A.D.F.; validation, M.M., A.D.F., Y.Z. and E.D.; formal analysis, M.M., A.D.F., Y.Z. and E.D.; investigation, M.M., A.D.F., Y.Z. and E.D.; resources, E.D.; data curation, M.M., A.D.F. and E.D.; writing—original draft preparation, M.M., A.D.F., and Y.Z.; writing—review and editing, M.M., A.D.F., Y.Z. and E.D.; visualization, A.D.F. and Y.Z.; supervision, E.D.; project administration, E.D.; funding acquisition, E.D. All authors have read and agreed to the published version of the manuscript.

**Funding:** This research was external funded by LKAB and the Swedish Energy Agency.

**Data Availability Statement:** Data that can be shared have been shown in the result chapter and Appendix C, other data is internal Epiroc and is unavailable due to privacy.

**Acknowledgments:** We would like to acknowledge Epiroc, who have provided mining machine data and test opportunities in their underground lab facilities, and the mining company Boliden site Kristineberg, where the initial test was carried out. Concludingly the mining company LKAB for including the research project SP13 'Monitoring of airflow and airborne particles, to provide early warning of irrespirable atmospheric conditions' within the academic program of the Sustainable Underground Mining (SUM) project. This academic program extends from year 2021 to 2024 and is joint financed by LKAB and the Swedish Energy Agency.

**Conflicts of Interest:** The authors declare no conflict of interest.

## Appendix A

Procedures for regular inspections and physical checks of vehicles and other equipment in underground mines relate to the inspections of hydraulic, fuel, and electrical systems to prevent underground fires [12]. Given the frequent fires, inspections are an obvious preventive activity to prevent causes of deviations such as abrasion, vibration, wear, and corrosion. The Fire Safety Committee of the Swedish Mining Industry's Health and Safety Committee [55] suggests an annual written fire inspection of vehicles and machines used in underground mines, providing a checklist and a guideline on the annual fire inspection. The Department of Mineral Resources in South Africa [56] defines regular inspections of working areas to monitor compliance with fire controls, including preventive procedures, a 250 h inspection of mobile equipment, as well as equipment pre- and post-maintenance inspections, as fire prevention tools. Detection of fires on mining machines is highly dependent on the operator or miners in the vicinity [57]. Utilizing their remarkably sensitive olfactory senses, early detection of smoke becomes feasible, enabling the identification of warning signals encompassing aberrant behavior, heightened vibrations, or unexpected shocks. In autonomous mining, the loss of these detection methods will have to be compensated and other means must be deployed. By detecting faulty equipment, such as overheated cables, motors, brakes, pressure changes on tires, oil leakage in engines or hydraulics, or similar in an early state fault, escalation of fires might be avoided. A capsized mining vehicle can produce a very high heat release rate (HRR) and temperature (°C) [5,14], which can cause rock movements so that part of the mine becomes unusable and causes long production stops until the rock wall has been secured again. In addition, it entails extensive remediation work and, in the worst case, a fatal outcome.

## Appendix B

The major fire-causing faults in the mining are highlighted below.

Engine overheating: Engine overheating in mining heavy-duty vehicles and machines can occur due to several potential hazards. These hazards include poor maintenance, coolant system problems, fan problems, exhaust system problems, operating conditions, fuel system problems, and electrical system problems. Lack of proper maintenance, clogged or blocked cooling passages, leaks, cracks, insufficient coolant flow, and malfunctioning cooling fans or exhaust systems are some common reasons for engine overheating. Regular maintenance, inspections, proper operator training, and monitoring of engine temperature gauges can help prevent engine overheating in mining trucks.

Cable overheating: Cable overheating in mining machines and heavy-duty vehicles can be a significant issue that needs attention to prevent equipment damage and electrical failures. Several factors contribute to cable overheating in these vehicles. Overloading the electrical system beyond the cable's rated capacity can cause excessive current flow, leading to overheating. Poor cable connections, such as loose, corroded, or damaged connectors, create resistance and heat buildup. Inadequate cable sizing, using undersized cables for high-power applications, increases resistance, and causes overheating. Damage to cable insulation, whether due to wear, mechanical stress, or environmental factors, can result in heat escape and localized overheating. Extreme ambient temperatures, inadequate

cooling, and overheating of nearby components can also impact cable temperature. Regular inspection, maintenance, and monitoring are essential to identify and address these causes promptly. Proper cable sizing, secure connections, intact insulation, and suitable environmental conditions should be ensured to mitigate cable overheating risks and ensure safe and reliable electrical operation in mining machines and heavy-duty vehicles.

Cable Reel Problems: Trailing cables layered on equipment cable reels without proper ventilation can overheat and cause insulation damage. The repeated winding and unwinding of cables can cause mechanical stress and cable fatigue. Additionally, voltage drop issues and difficulties in handling cables can arise. When electrical current passes through the conductors of a trailing cable, it generates heat. If this heat exceeds the capacity of the conductor insulation, cable outer jacket, and surrounding environment (such as the cable reel), it causes an increase in temperature for the conductor and other cable components. When the conductor temperature surpasses the insulation's normal operating temperature rating (typically 90 °C for modern trailing cables), the insulation's effectiveness starts to deteriorate. Prolonged exposure to temperatures beyond the insulation's rating can result in insulation breakdown or deterioration, significantly raising the risk of short circuits, fires, or even explosive energy releases. To prevent insulation degradation and potential hazards when using trailing cables on mining machinery, it is vital to focus on heat dissipation and effectively manage cable temperature. Implementing measures such as ensuring adequate ventilation, using cable reels with appropriate capacity, following correct winding and unwinding practices, and conducting regular inspections and maintenance are essential steps to mitigate these risks.

Brake leakage: Brake leakage is a potential hazard that can occur in mining heavy-duty vehicles and machines. The causes of brake leakage can be due to several reasons, such as wear and tear of the brake lines and hoses, corrosion or damage to the brake components, and improper installation or maintenance of the brake system. It can also be caused by overloading or overheating of the machinery, which can put a strain on the brake system, leading to leaks. To detect brake leakage, regular inspections of the brake system should be conducted. Operators and maintenance personnel should look for signs of leaks, such as fluid puddles, wet spots, or brake fluid loss. They should also check the brake lines, hoses, and fittings for damage, wear, or corrosion. Any issues found should be addressed immediately, and the brake system should be repaired or replaced as necessary. Proper maintenance and inspection of the brake system components can help prevent brake leakage and ensure the safe and efficient operation of mining heavy-duty vehicles and machines.

Oil leakage: Oil leaks in mining heavy-duty vehicles and machines can occur due to several potential factors, including mechanical wear and tear, corrosion, overheating, and other environmental factors. The harsh environments in which mining equipment operates subject it to high levels of stress and wear, causing components such as gaskets, seals, and hoses to degrade, crack, or fail. Exposure to corrosive substances, extreme temperatures, and humidity can also cause degradation of equipment components, leading to oil leaks. Operators and maintenance personnel can reduce the risk of oil leaks by regularly inspecting and maintaining mining equipment, using high-quality equipment components, and receiving proper training.

Tire wear: It refers to the gradual deterioration of the tire's tread and overall condition over time. Tire wear in mining machines and heavy-duty vehicles occurs due to several factors. The abrasive nature of mining sites, characterized by rough and uneven surfaces, rocks, gravel, and debris, leads to accelerated tire wear through abrasion. Overloading the vehicles beyond the tire's load capacity puts excessive stress on the tires, causing them to wear more quickly. Improper tire pressure, whether underinflation or overinflation, contributes to accelerated wear and uneven wear patterns. Misalignment of the wheels can cause tire scrubbing against the road surface, resulting in uneven wear. Aggressive driving techniques, such as hard braking, rapid acceleration, sharp turns, and skidding, accelerate tire wear. Neglecting proper tire maintenance, including regular rotation, inspections, and

timely replacement of worn-out tires, also contributes to increased wear. Managing these factors through proper maintenance, monitoring, and promptly addressing issues is crucial for minimizing tire wear and maximizing tire life. Given some experience, the probability of buying gas incertain fraud conditions can be obtained and listed in Table 1.

**Appendix C**

Example: Engine overheating detection module

Step 1: Identify the Variables and Define Their States

Variables:

- Engine temperature (absolute value) and $CH_x$ deviation (from normal values)

States;

- Engine temperature: OK (Below 90 °C), low (90 °C–105 °C), medium (105 °C–120 °C), and high (Above 120 °C);
- $CH_x$ deviation: OK (Below 0.1 ppm), low (0.1 ppm–0.7 ppm), medium (0.7 ppm–1.2 ppm), and high (Above 1.2 ppm).

Temperature and $CH_x$ are parameters that can characterize engine overheating in the mining sector. Their states were defined based on recommendations and experiences of mining machine operators in relation to engine overheating states.

Step 2: Construct the Bayesian Network (BN)

Parent node: Engine overheating (states: OK, low, medium, high)

The states for the parent node (engine overheating) are defined based on the nature of the problem and its relevance in the maintenance decision-making. In this case, "OK" represents normal operating conditions, while "low," "medium," and "high" represent increasing levels of engine overheating severity. The maintenance actions and readiness levels for engine overheating depend on the severity of the overheating condition. "Low" may involve minor maintenance and restricted operation, "medium" requires immediate attention and potential vehicle shutdown, while "high" demands immediate action and may result in the vehicle being inoperable until the issue is fully resolved to ensure safety and prevent fire hazards.

Child nodes:

- Engine temperature (states: OK, low, medium, high)
- $CH_x$ deviation (states: OK, low, medium, high)

Step 3: Determine Prior Probabilities

Prior probabilities are determined based on historical data and expert knowledge. Overheating can occur on different levels, while severe overheating is rare over the engine's lifetime. For this example, the following prior probabilities are considered:

- P(Engine Overheating = OK) = 0.99
- P(Engine Overheating = low) = 0.025
- P(Engine Overheating = medium) = 0.03
- P(Engine Overheating = high = 0.001

Step 4: Compute Conditional Probability Tables (CPTs)

The CPT of the engine overheating BN module is provided in Table 2, Section 2.4. The CPTs define the relationship between parent and child nodes. For instance, when the engine is under the 'OK' operating state, the probability of the engine temperature being in the "OK" state is 0.99. The remaining 0.1 probability value is dedicated to sensor noise. A very small $CH_x$ is expected at this operating condition of the engine. However, due to sensor errors, some $CH_x$ observations are expected to fall in the "M" zone.

Step 5: Infer the State of the System

To infer the state of engine overheating, we need new data on the engine temperature and $CH_x$ deviation. Assuming the following measurements:

- Engine temperature: 110 °C = "M" state
- $CH_x$ deviation: 0.8 ppm = "M" state

It is important to note that, for a given engine temperature within its "Medium" state, it is more likely that the $CH_x$ values fall within the same "Medium" state range due to the relationship they have.

Bayes' theorem and inference algorithms are also used to calculate the posterior probabilities for each state of engine overheating.

- P(Engine Overheating = OK | Engine Temperature = M, $CH_x$ Deviation = M)
- P(Engine Overheating = L | Engine Temperature = M, $CH_x$ Deviation = M)
- P(Engine Overheating = M | Engine Temperature = M, $CH_x$ Deviation = M)
- P(Engine Overheating = H | Engine Temperature = M, $CH_x$ Deviation = M)
- P(Engine Overheating = OK | M, M) $\propto$ P(Engine Overheating = OK) $\times$ P(Engine Temperature = M | OK) $\times$ P($CH_x$ Deviation = M | OK) $\propto$ 0.99 $\times$ 0.004 $\times$ 0.05 = 0.000198
- P(Engine Overheating = L | M, M) $\propto$ P(Engine Overheating = L) $\times$ P(Engine Temperature = M | L) $\times$ P($CH_x$ Deviation = M | L) $\propto$ 0.005 $\times$ 0.02 $\times$ 0.95 = 0.0000475
- P(Engine Overheating = M | M, M) $\propto$ P(Engine Overheating = M) $\times$ P(Engine Temperature = M | M) $\times$ P($CH_x$ Deviation = M | M) $\propto$ 0.004 $\times$ 0.97 $\times$ 0.95 = 0.01846
- P(Engine Overheating = H | M, M) $\propto$ P(Engine Overheating = H) $\times$ P(Engine Temperature = M | H) $\times$ P($CH_x$ Deviation = M | H) $\propto$ 0.001 $\times$ 0.01 $\times$ 0.99 = 0.000000099

Calculate the normalization constant (Z):

Normalization constant (Z) = sum of all unnormalized probabilities
Z = 0.000198 + 0.0000475 + 0.01846 + 0.000000099 $\approx$ 0.018705599

Now the probabilities are normalized:

- P(Engine Overheating = OK | M, M) $\approx$ 0.000198/0.018705599 $\approx$ 0.010597
- P(Engine Overheating = L | M, M) $\approx$ 0.0000475/0.018705599 $\approx$ 0.002540
- P(Engine Overheating = M | M, M) $\approx$ 0.01846/0.018705599 $\approx$ 0.986062
- P(Engine Overheating = H | M, M) $\approx$ 0.000000099/0.018705599 $\approx$ 0.000005

The state with the highest probability is "Engine Overheating = Medium" with a probability of approximately 0.986. Based on the observed data (Engine Temperature = Medium, $CH_x$ = Medium) and the estimated posterior probabilities, the decision by the BN is that the severity of engine overheating is "Medium."

The sensitivity of the analysis to the prior probability values has also been examined based on the following three cases:

Case 1:

- P(Engine Overheating = OK) = 0.98
- P(Engine Overheating = L) = 0.01
- P(Engine Overheating = M) = 0.008
- P(Engine Overheating = H) = 0.002

Case 2:

- P(Engine Overheating = OK) = 0.97
- P(Engine Overheating = L) = 0.015
- P(Engine Overheating = M) = 0.012
- P(Engine Overheating = H) = 0.003

Case 3:

- P(Engine Overheating = OK) = 0.95
- P(Engine Overheating = L) = 0.025
- P(Engine Overheating = M) = 0.02
- P(Engine Overheating = H) = 0.005

The estimated probabilities through the inferencing indicated that "Engine Overheating = Medium" is the most likely state.

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
