# Peer review of "Holistic Approach Promotes Failure Prevention of Smart Mining Machines Based on Bayesian Networks"

_machines, doi:10.3390/machines11100940_

Round 1

Reviewer 1 Report

This manuscript needs major improvements:

1.       The introduction should be refined to strengthen the literature review in a systematic way, with a focus on current research status, especially in the field of intelligent failure prevention. It is better to incorporate background section into the introduction section. Some contents in existing introduction and background sections may be moved to the Appendix.

2.       Notwithstanding the last paragraph of Section 2, the novelty of the work presented in this manuscript is unclear. It is necessary to clearly state what challenges were addressed, why they are challenges, and more importantly, what new knowledge/technology was added.

3.       Bayesian networks are well established and have been used widely. As an application of this method, the selection of node variables, the network structure, elements connectivity and the probability determination should be described in detail. Sensitivity analysis should also be performed to validate the probability parameters of the Bayesian network. The general procedure presented on Page 8-9 is not enough.

4.       Conclusion section needs to be revised to highlight the new findings/knowledge/technologies gained from this study.

5.       Poor writing quality. There are many unclear sentences and grammatical errors. Some paragraphs are difficult to follow.

Some more specific comments are se follows:

6.       Line 60-63: “Introducing new fleets of mining machines that are, for example, battery-powered, encounters patrols and challenges the mining industry as research in the field is vague, ensuring that productivity is maintained or further improved while workplace safety is not compromised.” This sentence is difficult to understand. What are the challenges?

7.       Line 69-98: This paragraph is unnecessarily long.

8.       Line 261-262: “This work uses the sensors signals as input to a decision tree system.” As the decision tree plays a critical role in the fire hazard detection system, some details of the decision tree should be provided, including the tree structure, parameters, and the connection between the BN and the decision tree.

9.       Line 262: “All this constitutes the gap which the study aims to bridge.” What is “this”? Why does it constitute a research gap?

10.   Line 271-272: “Another challenge was deciding where to mount the sensor, what hight in relation to the different gas densities.” What does “what hight” mean? How was this challenge solved?

11.   Line 272-274: “The aim of the mock-up test was to test...” The details of the mock-up test should be provided.

12.   Line 314-315: “These states are described linguistically as YES, NO, OK, Low (L), Medium (M), and High (H).” Examples should be provided to demonstrate how the states were classified.

13.   Line 317-318: “The prior probabilities of the variables were determined based on historical data and expert knowledge.” Examples should be provided to demonstrate how the prior probabilities were determined.

14.   Line 324-325: “The CPTs for the proposed structure were assigned based on historical data and expert 324 knowledge.” Examples should be provided to demonstrate how the CPTs were determined.

15.   Line 330: “Interpreting the probability values obtained through the inference process…” How were the probability values interpreted?

16.   Page 8: “The general procedure used to develop the detection modules corresponds to the following steps:” At least one example should be provided to demonstrate the application of this procedure in the module development.

17.   Line 362: “A positive observation, during the test was that the sensor was…” Which sensor?

18.   Line377-378: “Other signals can be created through simulation models.” What signals were simulated? How were these signals simulated? How were the simulations validated?

19.   Page 13: Table 1 and Table 2. How were the cases in Table 1 determined?  Why were only these combinations considered? The overall probability of case 3 in Table 2 is less than 100%, the same as case 7.

1.       Poor writing quality. There are many unclear sentences and grammatical errors. Some paragraphs are difficult to follow.

Author Response

Many thanks for the valuable work you all reviewers and editor have carried out by reading the article and for the excellent and valuable feedback we received. The article has been updated according to the feedback to our best interpretation. The sections have been marked with the following colours Reviewer 1, Reviewer 2. When the same feedback was repeated by more than one reviewer the text has been marked with several colours. Please see the response in the attached document.

Reviewer 2 Report

First of all, thank you for contributing to industrial maintenance, especially in the intelligent and autonomous paradigm of the upcoming era of artificial intelligence. I will start by saying that the manuscript has enormous potential and exciting ideas presented to the readers. However, I have extensive remarks and suggestions for improvement of the manuscript. I will give you some pointers and directions.

Abstract: Remove spaces in the abstract (if this is not a pdf error). The first sentence in the abstract, "Future autonomous mining processes from drilling to the harbour will demand new approaches for preventing hazardous situations for from occurring." Do not use "cannot" or regular slang in academic articles. Use capital letters to describe acronyms, such as Artificial Intelligence (AI).

Introduction: Although you've provided an interesting analogy, I would not recommend using the first paragraph in the introductory part of the manuscript. This a technical/engineering study. However, you may address the notion as you did in the 2nd paragraph, but sentences from 29 to 31 are a bit extreme; I think you would agree. In line numbers 48-49, you narratively state "percentages", while in line number 39, you express symbolically as %. I suggest you stick to the one or the other. In line numbers 63-66, you list risks associated with battery-power mining machines "...fire, smoke development, mitigation, and fire suppression, ventilation, evacuation and released gases..." I have trouble understanding mitigation, fire suppression, ventilation, and evacuation risks. Are they risks or countermeasures to fire-related consequences? Please make sure you've identified risks as probabilities leading to undesirable outcomes. Or at least, if I am mistaken, can you please explain what you mean by this?

Line numbers 77-78 "In addition, [11] analyzes of fires...", Do you mean analysis? I am writing suggestions for improvement in UK English, not to be mistaken. As such, I see that punctuation, commas, and other grammatical errors exist throughout the paper. Try using free softwares (e.g., Grammarly, Word Checker, etc.) to remove these errors. Line numbers 91-92 "An interesting observation is that, e.g., machines produced by manufacturers for above ground, such as loaders..." This sentence is not clear and concise and should be improved. Also, if you are talking about "underground" and "above ground" mining, please use the terminology of open pit and close pit mining, respectively. Do not use, e.g., use "for instance, for example, such as etc.," in the middle of the sentence. Line numbers 93-94, "An example concerns the speed, machines...." try to reformulate it. "Speed restrictions to closed-pit mining machines of 18 km/h according to standard XX does not apply to open-pit mining machines...". 

In general, the introduction should be extensively reformulated and rewritten. This form resembles more a thesis structure than a scientific article. I would kindly ask you to reduce the introduction or merge 1. Introduction and 2. Background into a single section. If you merge, add subsections 1.1 Background, 1.2 Rationale, 1.3 Aims and Objectives. Also, I do not see a clearly stated objective and aims of the study. Instead, just implicit remarks about the direction of the research study have gone. What specific issues are you addressing? What has been done similarly? What challenges or research gaps are you addressing? Please try to be concise.

In line sentences, 115-116, do not use "preventative" if you want to state preventive actions/activities/procedures. This term is not used often if you are practising and writing about the industrial maintenance domain.

In line sentences, 119-120, do not use "With the help of their noses..." Rephrase the term to smell or similar.

In 2.2, paragraphs should be justified. Considering that you are already writing in a way that is the conclusion "Future mining will need to be smart..." consider writing in the present tense and active voice. You are writing for the reader as this is a scientific article, not a novel piece. Do not repeat the explanation of acronyms AI. Also, sentences from 160-167 are highly flawed. Try using definitions of corrective, preventive and predictive maintenance. The research is far from this term to sustainable, data-driven, energy-based maintenance. Stating that "Corrective maintenance where maintenance is carried out when something goes wrong..." is not in line with academic writing. Please improve your academic expression. Please work with mentors or someone from academia. 

There are much more suggestions for improvement. Therefore I will stop at the water's edge. Although the topic is exciting and could make a potential impact since the data is essential in this area, I strongly recommend writing an entirely new article from the start. The article has serious flaws, and unfortunately, I would not recommend that article in this form should not be accepted. Please, do not be discouraged, and do not take it the wrong way, but you need to increase your competence in writing and elaborating your findings more scientifically and academically. I hope this will motivate you and provoke you to improve yourself.

All the best.

Extensive editing is required.

Author Response

Many thanks for the valuable work you all reviewers and editor have carried out by reading the article and for the excellent and valuable feedback we received. The article has been updated according to the feedback to our best interpretation. The sections have been marked with the following colours Reviewer 1, Reviewer 2. When the same feedback was repeated by more than one reviewer the text has been marked with several colours. Please see attached document, page 2. 

Round 2

Reviewer 1 Report

1.       The responses to the reviewers’ comments are too simple. The reviewers’ comments should be answered point by point.

2.       There are still many unclear sentences and grammatical errors. The structure of the revised manuscript is still not well organized.

3.       Line 497: “the BN model achieved a 98.9% accuracy in detecting such leakages”. How was the accuracy determined?

There are still many unclear sentences and grammatical errors. 

Author Response

1000 Thanks for the feedback, please see the attachement.

Reviewer 2 Report

Dear authors,

I am glad you've edited and improved the article.

However, I also have the following suggestions for improvement. First please follow instructions and guidelines for authors since referencing and equations used in the article for citation and referencing are not done accordingly. This also holds for tables.

Considering the manuscript, it is difficult to focus on the text since a lot of things are marked, I suppose by the review feature in Word. Nevertheless, from what I've read so far, I would suggest extending the literature a bit more. For instance, it would be good to support the argument of hydraulic leakage. This is in fact argued that leakage and pipe (hose) bursting are some of the most common causes of hydraulic failures (see https://link.springer.com/chapter/10.1007/978-3-030-88465-9_62). As for the BN, I would suggest the usage of JASP, which is an open-source software developed by the University of Amsterdam (*if I am not mistaken*). I believe you have a really easy way of using BN complete analysis and graphical representation of the network structure. This may help improve the quality and insight of figures and the analysis. Note: I do not require all of this as a must, but it would strongly improve the quality and validity of your findings and results. Also, based on the Bayesian network analysis in JASP, there is a really nice representation of the association between the nodes, while also a nice representation of the edge weights. You can just use BN throughout the whole paper, you do not have to mention and explain the acronym several times in the paper, which is fine. Also, please check the figure sequence, I see that after Figure 4/5 is Figure 7, or maybe it is a mistake on my laptop. The Figure 8 is not visible on my screen. Please check that...Also, please use the equation editor in Word do not write equations as text (see eq.3). Also, use the appropriate reference style for formulas from the Style editor in Word. I believe, if you used a machine template, that there is already an equation style in the editor.

When you edit next time, please send separate versions of the revised article and without revision (correction), because it will be much easier to understand and follow up the article.

Minor editing is required. I would suggest to the authors to use free software trial versions like Grammarly to correct spelling mistakes.

Author Response

1000 Thanks for the feedback, please see the attachment.

Round 3

Reviewer 1 Report

This manuscript still needs major improvements:

1.       The quality of writing still needs to be improved. A few examples are as follows:

(1)    “Studies concerning mining fires [10] for metal and non-metal mining shows that the…” (Line 94-95)

(2)    “…reason both what causes the error but also what measures are required to prevent something from occurring.” (Line 125-126)

(3)    “A study that was conducted in Mexico [20] the following symptoms on miners 183 could be observed: shortness…” (Line 183-184)

(4)    “In a scenario where a source for instance an overheated cable near a rock face, emits gases.” (Line 445-446)

(5)    Figure 6 and Figure 9: units are missing.

2.       The information added in Section 2.4 is insufficient: (1) often only the variables, states and values used/selected are described without explaining why and how they were determined; (2) no examples are provided for the last two steps (Infer the state of the system and Test the model). It is best to demonstrate the entire procedure (used to develop the detection modules) based on a single hazard, such as engine overheating. Why should the engine overheating be monitored through measuring engine temperature and CHx? Why should the engine temperature be described by 4 states, but CHx be described by 3 states? How were the relevant probabilities determined? The data used for the analysis should also be provided. Although it is mentioned that some data were from reference [26], [26] is a conference paper and is difficult for readers to access.

3.       Line 483-485: “The estimated probability values presented in Table 3, indicate the overall detection accuracy of the proposed BN scheme.”. Why?

4.       Line 488-494: More details should be provided to explain how these states were determined. For example, why are there only two cases for the hydraulic leakage detection BN model and the Brake leakage detection BN model? Are these two variables perfectly related?

5.       Line 503-504: “If temperature is the only measurement available, the same temperature can be serious for one type of cable insulation while not for another, as shown in case 5 to 8 of Table 3.” Table 3 does not show that “the same temperature can be serious for one type of cable insulation while not for another”.

6.       Figure 10: It shows that the oil-lable (label?) and CH are both in "OK" status for the hydraulic system, but the probability of leakage is estimated to be very high. Why? Furthermore, were the estimated probabilities of BNs verified?

The quality of writing still needs to be improved. 

Author Response

Please see attached reviewer letter

Reviewer 2 Report

I'm fine with the corrections, please submit for publishing.

No major english check requried.

Author Response

Thanks for your great feedback, since reviewer 1 had additional request the attached file shows our actions.

Round 4

Reviewer 1 Report

This manuscript still needs major improvements:

1.       The quality of writing still needs to be improved:

(1)    Carefully check and improve not only the examples given by the reviewers, but also the entire manuscript.

(2)    Pay attention to the newly added content, such as “… it is essential to possess the capability to not only detect errors but also to comprehend their underlying causes …”.

(3)    Figure 6 and Figure 9: Axis titles and their units should be added directly to the figure. Figure 6: why is only the secondary axis is explained?  Figure 9: the timestep should have a unit such as second. “The values are scaled”. What does it mean?

2.       As an application of Bayesian Network, a complete example based on a single hazard (engine overheating, cable overheating or other hazard) must be provided to demonstrate the entire procedure for developing the detection modules from identifying the variables and defining their states until testing the Model. In this example, the following questions should be answered based on detailed data analysis:

(1)     Why and how were the variables selected?

(2)     Why and how were the states of each variables determined?

(3)    How were the relevant probabilities determined and verified?

(4)    How was the model verified?

3.       Line 355: “…the BN models will be tested on a new dataset from…”  Have the BN models tested?

The quality of writing still needs to be improved.

Author Response

Please see attached reviewer letter.
